# Study on Optimization of Mapping Method for Multi-Layer Cage Chicken House Environment

**DOI:** 10.3390/s25092822

**Published:** 2025-04-30

**Authors:** Zhaobo Zhang, Yanwei Yuan, Xin Dong, Yulong Yuan, Sa An, Yue Cao, Yang Li, Yuefeng Chen

**Affiliations:** 1State Key Laboratory of Agricultural Equipment Technology, Beijing 100083, China; 2Chinese Academy of Agricultural Mechanization Sciences Group Co., Ltd., Beijing 100083, China; 3Zhejiang Branch of Chinese Academy of Agricultural Mechanization Sciences Group Co., Ltd., Hangzhou 030006, China

**Keywords:** chicken house breeding, autonomous navigation, mapping method, point cloud matching

## Abstract

**Highlights:**

This article proposes an improved gmapping algorithm by combining an optimized ICP point cloud matching algorithm and a classification resampling method, which solves the problems of the insufficient number of effective particles, high repetition rate of environmental map information particles, and penetration phenomenon in the process of constructing maps in multi-layer cage architecture chicken coops.

**What are the main findings?**

**What is the implication of the main finding?**

**Abstract:**

This study delves into the mapping method for the navigation system of a chicken coop disinfection robot. It systematically analyzes the problems of insufficient effective particle count, high particle repetition rate in environmental map information, and penetration phenomenon in traditional SLAM laser point cloud mapping technology in chicken coop environments containing multiple layers of chicken cages. To address these issues, this paper proposes an optimized mapping method based on an improved ICP algorithm, significantly improving the laser point clouds’ registration performance. At the same time, by limiting the sampling of environmental map information particles within a specific range and optimizing the screening based on the predicted distribution of particle poses and the matching degree of the map, the diversity of particles and the accuracy of map information have been effectively improved. The field experiment results show that the maximum error of this method on the chicken coop environment map does not exceed 3.5 cm. The environmental characteristics of the chicken coop are maximally preserved, which verifies the effectiveness and robustness of this method and provides a scientific basis for the mapping method of the livestock and poultry breeding robot navigation system.

## 1. Introduction

As one of the core technologies in mobile robot research, simultaneous localization and mapping (SLAM) plays an important role in realizing autonomous navigation and environmental awareness [1]. The use of SLAM technology is a method that can build an environment map and realize robot positioning in an unknown environment in real-time by integrating the robot’s own perceptual data and motion information. It provides key technical support for the robot’s path planning, navigation, and task execution [2,3]. SLAM technology has an increasingly broad application prospect in autonomous driving, service robots, industrial automation, and so on, and it has become one of the hot topics in robot research.

The early SLAM research is mainly based on the extended Kalman filter method, which realizes the robot state estimation by linearizing the nonlinear system. However, the early SLAM algorithm is highly complex and challenging when dealing with complex environments. To overcome these limitations, Murphy and Doucet proposed a SLAM method based on RBPF, which decomposed the SLAM problem into two relatively independent problems: robot positioning and map construction [4], significantly reducing the algorithm’s complexity. The proposed FastSLAM algorithm [5] combines a particle filter and an extended Kalman filter to estimate robot attitude accurately. The FastSLAM algorithm effectively reduces computational complexity in more complex environments. However, the traditional particle filter method is prone to particle degradation in the resampling process, which leads to a decline in estimation accuracy. Grisetti et al. proposed a selective resampling method in 2005 to reduce particle dissipation by optimizing the resampling strategy to alleviate particle degradation effectively. However, while improving the estimation accuracy, this method may also lead to insufficient particle diversity and affect the algorithm’s robustness [6]. In 2024, the Wu Z.L. team proposed a cage chicken house egg detection method based on a classification weighting mechanism, which can improve the accuracy of data matching and data filtering [7]. In recent years, the improved resampling method has been one of the hot issues in the SLAM research field. However, it is not easy to fully meet the needs of high-precision map construction only by optimizing the resampling method, and other optimization strategies need to be combined to improve the algorithm’s performance further.

In constructing environmental maps, point cloud matching technology plays a key role. Besl et al. proposed Iterative Closest Point (ICP), which achieves pose estimation by minimizing the distance between adjacent frame point clouds [8]. However, traditional ICP algorithms have problems such as considerable computational complexity and significant cumulative errors [9,10]. Serafin et al.’s research shows that point cloud-based registration methods are prone to gradually accumulating errors when constructing local environmental models [11], primarily when registering adjacent observation data, which may lead to estimation drift and significant deviation from the actual environment. To reduce this impact, researchers have proposed an incremental point cloud matching method, which dynamically updates the reference model during each registration process to reduce drift errors. Although this method effectively improves registration accuracy, the computational efficiency of the algorithm will significantly decrease as the number of point clouds in the model increases linearly.

This paper proposes an improved robot mapping method to address the limitations of traditional SLAM methods. In response to the problem of excessive computational complexity in traditional ICP algorithms, when searching for the nearest corresponding point from the source point cloud to the target point cloud, this paper adopts an improved ICP algorithm to register laser point clouds precisely. By optimizing the point cloud matching strategy, the computational complexity is significantly reduced while improving the efficiency of extracting adequate point cloud information. In response to the problem of insufficient particle diversity in traditional gmapping algorithms, this paper proposes an improved resampling method that effectively improves particle sets’ diversity and enhances the algorithm’s robustness by weighted classification resampling of particles [12]. We compared and analyzed the parameters of three types of LiDAR, and Table 1 shows the comparison results of LiDAR. The results indicate that the measuring distance range of Mid-360 LiDAR is 40 m, which is suitable for short-distance mapping experiments in chicken coop environments. The Mid-360 LiDAR adopts a cyclic climbing point cloud scanning method with a horizontal field of view angle of 360° and a vertical field of view angle of 59°. Compared to the other two types of radar with a limited field of view and parallel point cloud scanning, the Mid-360 LiDAR provides more comprehensive information on the chicken coop environment. Therefore, we conducted on-site mapping experiments using the Mid-360 LiDAR sensor (RoboSense, Shenzhen, China) and comprehensively evaluated the algorithm performance to verify the accuracy and stability of the method.

The research work of this paper has important theoretical significance and practical application value. At the theoretical level, the improved gmapping algorithm provides a new idea for solving the problems of particle degradation and point cloud registration accuracy in SLAM [13,14]. At the application level, the optimized mapping method can provide more reliable technical support for the autonomous navigation of mobile robots in complex environments. Through the theoretical analysis and experimental verification of the system, the improved method proposed in this paper ensures the accuracy of map building and significantly improves the algorithm’s robustness [15].

## 2. Materials and Methods

In the chicken house breeding environment, given the special conditions of narrow space and great difficulty for robots to travel, this study adopted the integrated design method to design a chassis structure of the steering wheel by the driven wheel, which is suitable for the high-load bearing disinfection system. Figure 1 shows the chassis structure model designed in this study. The dimensions of the chassis structure are 493 × 500 mm, which have the characteristics of small volume, lightweight, and strong adaptability. It can meet the needs of omnidirectional movement and effectively reduce the damage to the core components of the chassis caused by accidental impacts during driving.

The walking control hardware system adopts a modular design method, and the entire system consists of three main parts: the central control module, the autonomous navigation module, and the walking control module. The autonomous navigation module is responsible for collecting point cloud information and robot chassis pose information. The main control module processes these point cloud data, constructs high-precision environmental maps, and implements autonomous navigation functions. The walking control module receives the optimal control strategy sent by the central control module, achieving precise motor speed control and ensuring that the chassis can walk autonomously. Figure 2 shows the hardware control system designed in this study, which meets the unique requirements of the chicken coop breeding environment and achieves autonomous and safe walking of the chassis through modular design and precise control algorithms.

The gmapping algorithm is a particle filter-based SLAM method proposed by Grisetti, which improves some of the problems of the RBPF algorithm and has been widely applied in robotics. The RBPF algorithm decomposes the SLAM problem into two problems: one is the robot localization problem, and the other is the map-building problem. Based on the particle filtering method, the RBPF algorithm can extract key information from sensor and odometer data, accurately estimating the map and robot pose [16].

During the resampling process, the RBPF algorithm may experience particle degradation, where some particles have excessively high weights while others have very low weights and may even be filtered out. This results in particle aggregation in some areas. In contrast, the number of particles in other areas significantly decreases, greatly reducing the diversity of particles and affecting the accuracy of robot pose state estimation. Based on improving RBPF, the gmapping algorithm introduces an adequate particle number Neff parameter.(1)$$N_{eff}=\frac{1}{\sum_{i=1}^{N}{\left(w_{i}^{i}\right)}^{2}}$$

When the number of effective particles is less than the threshold, resampling is performed to filter out particles with lower weights while copying particles with higher weights. This method can significantly reduce the number of resampling operations in RBPF and effectively solve the problem of particle degradation. Because particles with significant weight carry environmental information closer to the actual environment, particle filters will screen particles with different weights, significantly reducing computational complexity while preserving environmental information. After multiple resampling, the number of high-weight particles will far exceed that of low-weight particles [17,18]. During resampling, particle filtering may remove particles with moderate weights containing information closer to the actual environment. The convergence of particle swarm optimization results in a significant deviation between the estimated pose state of the robot and the accurate information. In addition, there is noise during the robot’s motion, and the laser sensor is also affected by the noise, which ultimately leads to a particular deviation between the 2D grid map constructed by the robot and the actual scene.

In response to the problem of differences between map construction and real environments in the gmapping algorithm, this paper conducts optimization research on point cloud matching methods and resampling methods to optimize the gmapping algorithm. In response to the measurement errors and observation noise in LiDAR scanning of point clouds, this paper uses an improved ICP algorithm to perform point cloud matching on point cloud data, achieving precise registration of point cloud data. Optimization research on resampling methods aims to avoid particle filtering out particles highly similar to accurate environmental information and increase particle diversity.

LiDAR has high-precision point cloud scanning data, and using laser odometry for frame-to-frame matching is one of the important techniques in traditional SLAM algorithms [19]. On a mobile robot with a fixed LiDAR, there is a fixed positional relationship between the LiDAR and the mobile robot, and the position changes of adjacent point clouds are mainly affected by the pose changes of the LiDAR. By determining the positional change relationship between adjacent frame point clouds, the pose transformation relationship of the moving robot at adjacent times can be obtained, achieving the pose transformation of the moving robot at adjacent times. The traditional ICP algorithm is a matching method based on inter-frame point clouds for 3D surface matching. It matches the point clouds of adjacent frames point-to-point to obtain the Euclidean rotation translation matrix and iterates repeatedly until the error is below the threshold.

The ICP algorithm is a key technology for point cloud data registration, and it can also be used for 2D radar point cloud matching to improve the registration accuracy of point clouds [20]. The presence of outliers or noise in point cloud data will significantly reduce the accuracy of point cloud matching. Finding the nearest neighbors between the source point cloud and the target point cloud requires much computation, especially when the number of point clouds is enormous, significantly reducing the computing speed. In addition, conventional ICP methods are sensitive to the initial matching results. When the initial matching results are not ideal, the number of iterations will continue to increase, resulting in increased computational complexity and consuming a large amount of computing resources.

The traditional ICP algorithm has poor robustness to data noise and outliers, which can easily lead to decreased registration accuracy. In response to the above issues, this article uses an improved point-to-point ICP algorithm to change the traditional ICP algorithm from using the distance between the two points closest to the Euclidean distance as the registration error function measure to the distance between the point in the source point cloud and the point closest to the target point cloud and the point connecting line, in order to improve matching accuracy and convergence speed.

The principle of improving the ICP algorithm is to set n_i_ as the normal of the line segment projected on the reference target surface Q_ref_, and, for the given source point cloud P and target point cloud Q, minimize the distance projected onto the reference surface Q_ref_ by rotating and translating p_i_, as shown in Figure 3:(2)$$\underset{q_{k+1}}{min}\sum {\left(n_{i}\left[p_{i}⊕q_{k+1}-\prod \left\{Q^{ref},p_{i}⊕q_{k}\right\}\right]\right)}^{2}$$

In the formula, ⊕ is the rotation translation symbol, p_i_ ⊕ q_k_ is the reference frame at time k, and the target frame after rotation translation transformation at time k+1 is p_i_ ⊕ q_k+1_.

The improved ICP method process is as follows:

(1) The matching accuracy is set before matching, and the error function is improved into the distance from the point to the straight line, that is, the distance from the point of the source point cloud to the line segment formed by the nearest two points of the target point cloud.

(2) Find two points, q_j1_^i^ and q_j2_^i^, closest to the source point cloud p_i_ in the reference target point cloud, and the matching line segment corresponding to the line connecting these two points is j_1_^i^–j_2_^i^.

(3) Set the square of the distance from the point of the source point cloud to the matching line segment as the error equation, where q_j_ is the point to be matched in the target point cloud.(3)$$E\left[R_{k},T_{k}\right]=\frac{1}{n}\sum_{i=1}^{n}{\left(\left[R_{k}p_{i}+T_{k}-q_{j}\right]n_{i}^{T}\right)}^{2}$$

(4) Calculate the transformation parameters R_k_ and T_k_ in the error equation to minimize the error function for subsequent iteration.

(5) Translate the point cloud p_i_ to p_i_^t^:(4)$$p_{i}=p_{i}⊕q_{j}=R_{k}p_{i}+T_{k}$$

(6) Calculate the average distance between p_i_^t^ and q_j_:(5)$$d=\frac{1}{n}\sum_{i=1}^{n}{∥p_{i}^{\prime }-q_{j}∥}^{2}$$

(7) When d is larger than the set matching accuracy, continue to match the latest point cloud data scanned by the LiDAR as the new source point cloud; otherwise, end the iteration.

The resampling operation of the traditional mapping method is aimed at a situation where the number of effective particles is low. In the resampling process, due to the increase in the number of repetitions, the difference between particles is getting smaller and smaller, which affects the accuracy of robot state estimation based on particle swarm optimization. In order to ensure the diversity of the particle swarm, the particles are divided into high-weight, medium-weight, and low-weight particles by weight [21]. Then, the medium-weight and low-weight particles are resampled on this basis. High-weight particles have better state estimation accuracy and are retained directly without resampling [22]. The information on medium-weight particles has a certain degree of credibility, which is retained and resampled to improve the diversity of particles. A significant deviation exists between the information carried by low-weight particles and the actual state. These low-weight particles are filtered out to improve the accuracy of robot state estimation. Calculate the average normalized weight of all particles based on the updated particle weight of the observation model:(6)$$\bar{\omega }=\frac{1}{N}\sum_{j=1}^{N}\omega_{i}^{\prime }$$

In this model, N is the number of particles and w_i_′ is the normalized weight. The particles are divided into three categories: high-weight, medium-weight, and low-weight. Among them, the weight of high-weight particles is greater than or equal to 2$\bar{w}$, the weight of low-weight particles is less than $\frac{\bar{w}}{2}$, and the weight of medium-weight particles is between the two. We copy the high-weight particles N_wi_ times. For the low-weight particles, we filter them directly. For the medium-weight particles, resampling is performed with a certain probability. For the ith medium weight particle, we resample with probability r_i_ and generate a random number. The random number generator generates the random number; its range is between 0 and 1, and the value generated each time is independent and evenly distributed [23]. This way, it can ensure that the resampling process is random and maintains a specific diversity. When r_i_ is close to 1, the probability of the particle being selected is high. When r_i_ is close to 0, the probability of the particle being filtered is higher. After resampling, a new set of particles is obtained for the next iteration update.

The particles are divided into three categories according to weight, which ensures the number of high-weight particles and resamples the particles with medium weight, avoiding the degradation of filtering performance caused by the unbalanced weight [24,25]. Because the particles with high weight will be copied many times, this method can reduce the sampling error, improve particle diversity, and enhance the accuracy and reliability of attitude estimation [26]. Classified resampling can more accurately reflect the posterior probability distribution and improve the estimation accuracy and robustness of the SLAM algorithm [27,28].

This paper optimizes point cloud matching and particle resampling methods using the traditional gmapping algorithm. The improved gmapping algorithm process is shown in Figure 4, and the specific process is as follows:(1)Initialize particles: Initialize the weights of all particles.(2)Data collection: Utilize the sensors installed on the robot to collect data on the environment, as well as the position and status of the robot.(3)Point cloud matching: Based on the collection of point cloud data by LiDAR, an improved ICP algorithm is used to match the point cloud data, achieving an accurate estimation of robot posture.(4)Sampling and Updating Weights: Updating the motion pattern of point cloud matching for each particle and calculating its weight.(5)Resampling: Based on this, classification resampling operations are performed, retaining particles with higher weights, filtering out particles with lower weights, and resampling particles with medium weights.(6)Update map: Combine the state of retained particles with the data observed by LiDAR to update the environmental map.

## 3. Results

Using the existing robot chassis experimental platform, equipped with LiDAR, and utilizing laser point cloud and image vision technology, a 2D grid map suitable for the autonomous navigation of robots is constructed using both the traditional gmapping algorithm and the improved gmapping algorithm proposed in this paper. Select indoor corridors, single cage passages in chicken coops, and complete environments in chicken coops for mapping experiments. Place the experimental platform at the starting position to be collected, plan the path of the collected environmental map, control the robot to move slowly and steadily on the ground, record radar point cloud data, and construct the required 2D grid map.

In the indoor corridor environment mapping experiment, due to the independent indoor enclosed environment, a closed-loop path was used to collect point cloud data, keeping the robot chassis in the middle of the corridor and constructing a complete grid map of the indoor corridor; in the experiment of mapping the channel between a single row of cages in a chicken coop, due to the straight path, a reciprocating path was used to achieve two point cloud data collections, maximizing the preservation of cage and environmental features on both sides, and constructing a grid map of the channel between a single row of cages; in the experiment of mapping the complete environment of the chicken coop, a reciprocating loop path was used to collect point cloud data, maximizing the preservation of the complete environmental characteristics of the chicken coop, including complex extraordinary obstacles, cage architecture, and other intelligent devices, to construct a complete grid map of the chicken coop environment.

The indoor long corridor experimental environment is shown in Figure 5. During the map construction task, the SLAM mapping algorithm controls the robot’s movement and obtains the structure and feature information of the indoor long corridor environment. Figure 6 shows the corridor environment map constructed by the traditional gmapping algorithm. The traditional gmapping algorithm accurately creates the features of the doors of the rooms in the corridor. Still, the constructed map has dense interference points, which do not match the actual flat corridor environment. The indoor long corridor environment map constructed by the improved gmapping algorithm can more accurately present the straight features of the indoor long corridor. Figure 7 shows the long corridor environment map constructed by the improved gmapping algorithm. The map construction effect of the door and the small room is better, and the noise is significantly reduced. The small obstacle features are also more apparent. In indoor extended corridor environments, the improved gmapping algorithm produces higher-quality maps than traditional gmapping algorithms.

Compared with the indoor long corridor environment, the actual environment of the chicken coop is more extensive, with numerous obstacles and complex structural features. In the actual environment of the chicken coop, traditional and improved gmapping algorithms are used to construct maps for individual cage channels and the entire chicken coop. Figure 8 shows the chicken coop’s actual environment and the mobile robot’s situation. Figure 9 shows a single row of inter-cage channel maps constructed by the traditional mapping algorithm. The maps constructed by the traditional gmapping algorithm exhibit noticeable map distortion and penetration phenomena, resulting in unclear boundaries and severe loss of environmental features. Figure 10 shows a 2D grid map of a single row of inter-cage channels constructed using an improved gmapping algorithm. The map boundaries are clear and accurate, and the complete environmental features can be preserved without map distortion.

In the actual environment of the chicken coop, traditional and improved gmapping algorithms are used to construct maps for the overall chicken coop environment. Figure 11 shows a complete chicken coop environment map constructed using the traditional gmapping algorithm. The map constructed using the traditional gmapping algorithm exhibits noticeable map distortion and penetration phenomena, and it is challenging to construct a complete chicken coop environment map due to excessive computational complexity. Figure 12 shows the complete 2D grid map of the chicken coop environment constructed by the improved gmapping algorithm, which can fully preserve the characteristics of the chicken coop environment without map distortion. The constructed map is complete and precise, providing a solid foundation for subsequent autonomous navigation technology. Compared with traditional gmapping algorithms, the improved gmapping algorithm can effectively solve map distortion and offset problems, generating more accurate and precise maps.

To quantitatively analyze the performance of the improved gmapping algorithm, we designed a comparative measurement experiment based on two methods for constructing maps. We selected three representative length indicators from the chicken coop environment, including the length of the coop, the distance between cages, and the length of cages. The length of the chicken coop is 98.00 m, the distance between the cages is 0.82 M, and the actual length of the cages is 91.30 m. We select the maximum measurement error as the evaluation index for algorithm accuracy and the mean square error as the evaluation index for algorithm stability. In the figure, the horizontal axis represents the number of measurements, the straight line represents the actual length data of the chicken coop scene, the solid line represents the length data measured by the traditional gmapping algorithm constructed map, and the dashed line represents the length data measured by the improved gmapping algorithm constructed map. The experimental data for measuring the length of the chicken coop are shown in Table 2, and the experimental results are shown in Figure 13.

The maximum error of the chicken coop length measured by the traditional gmapping algorithm is 9.0 cm with a mean square error of 0.0053. In comparison, the maximum error of the chicken coop length measured by the improved gmapping algorithm is 3.4 cm with a mean square error of 0.0004.

The experimental data for measuring the distance between cages are shown in Table 3, and the experimental results are shown in Figure 14.

The maximum distance error between chicken houses measured by the traditional gmapping algorithm is 5.2 cm, with a mean square error of 0.0015. The maximum error of the chicken house length measured by the improved gmapping algorithm is 2.5 cm, with a mean square error of 0.0003.

The experimental data for measuring the length of the cage are shown in Table 4, and the experimental results are shown in Figure 15.

The maximum error of the chicken coop length measured by the traditional gmapping algorithm is 10.5 cm, with a mean square error of 0.0050. The maximum error of the chicken coop length measured by the improved gmapping algorithm is 3.4 cm, with a mean square error of 0.0007. In three sets of measurement experiments, the maximum error of the measurement results of the improved gmapping algorithm is smaller than that of the traditional gmapping algorithm, and the mean square error is closer to 0. Therefore, the enhanced gmapping algorithm has stronger accuracy and higher stability.

## 4. Discussion

In this study, the mapping method used by the navigation system of a chicken house disinfection robot was deeply optimized to achieve the goal of building a complete and high-precision environment map. Based on the traditional gmapping algorithm, this study combines the working characteristics of SLAM LiDAR to propose an improved ICP algorithm. This method optimizes the resampling process to form an optimized gmapping mapping algorithm, effectively improving the accuracy and reliability of map construction.

The traditional gmapping algorithm can effectively handle the localization and map construction problems of robots when building environmental maps, but it is prone to map distortion and offset when facing complex environments. To address this issue, this study optimized the ICP algorithm to better adapt to the unique environment of chicken coops, particularly exhibiting stronger robustness when dealing with complex obstacles and environmental noise. In response to the problem of insufficient particle diversity in the resampling process of traditional gmapping algorithms, this study proposes a new resampling strategy that divides particles into three categories: high-weight heavy particles, medium-weight particles, and low-weight particles. During the resampling process, high-weight particles were retained, low-weight particles were filtered out, and medium-weight particles were resampled to increase particle diversity, effectively avoid particle degradation problems, and improve the stability and accuracy of map construction. The improved ICP algorithm replaces the traditional ICP algorithm, which uses the Euclidean distance between the nearest two points as the matching error function metric, with the distance between the two closest points in the source point cloud and the target point cloud, increasing the convergence speed. The classification resampling method selects more effective particles while limiting the total number of particles, reducing computational complexity. The experimental results show that the optimized gmapping mapping algorithm can significantly reduce mapping errors, with a maximum error of only 3.4 cm. In the on-site experiment of the chicken coop, the maximum error in measuring the length of the coop was only 3.4 cm, and the maximum error in the spacing between chicken cages was only 2.5 cm. The maximum error in the length of chicken cages was only 3.4 cm, meeting the requirements for the unmanned operation of the chicken coop disinfection robot.

## Figures and Tables

**Figure 1 sensors-25-02822-f001:**
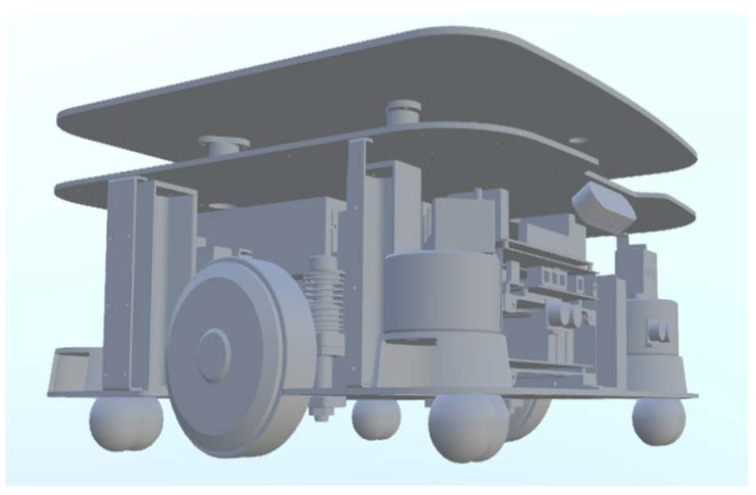
Chassis structure model.

**Figure 2 sensors-25-02822-f002:**
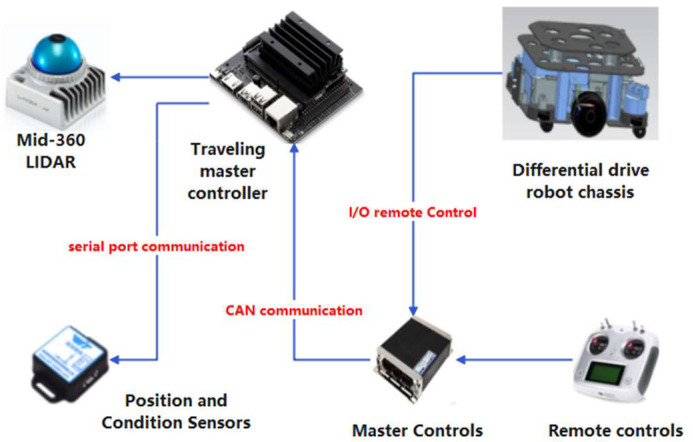
Hardware system design.

**Figure 3 sensors-25-02822-f003:**
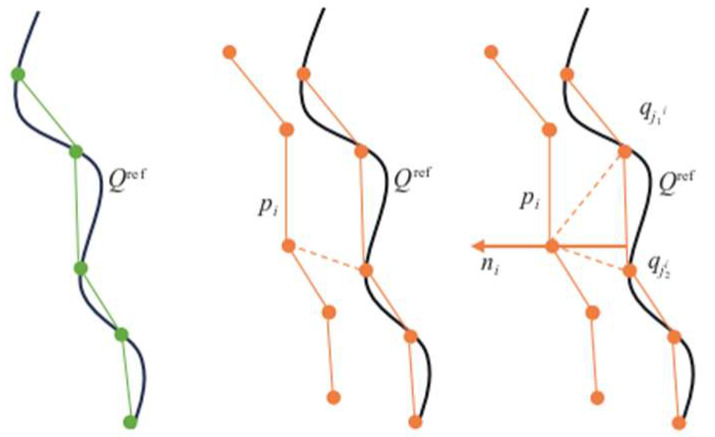
Improved distance from ICP point to straight line.

**Figure 4 sensors-25-02822-f004:**
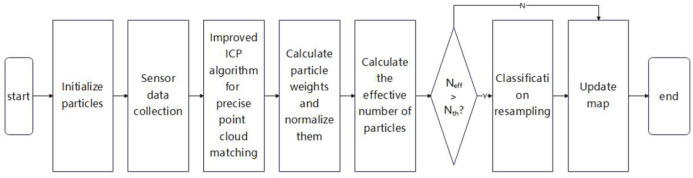
Flowchart of improved gmapping algorithm.

**Figure 5 sensors-25-02822-f005:**
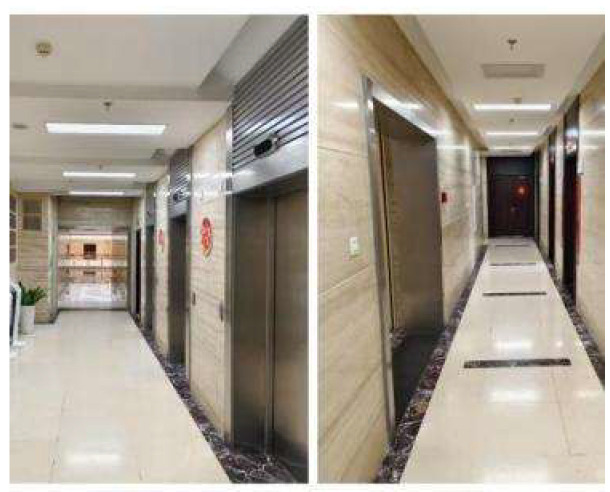
Indoor long corridor environment.

**Figure 6 sensors-25-02822-f006:**
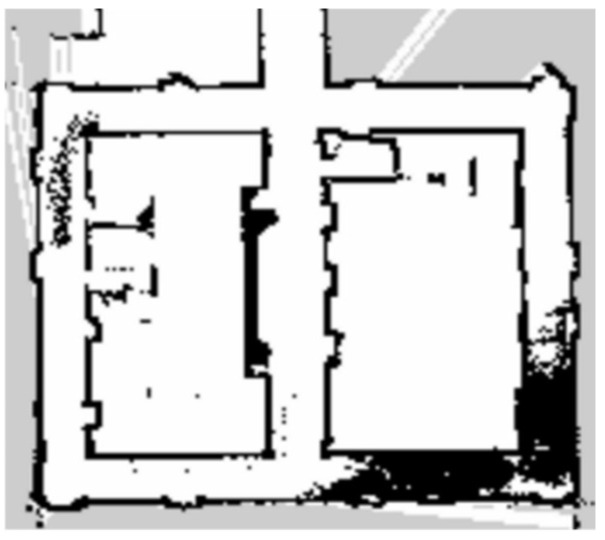
Indoor long corridor map constructed by traditional gmappig algorithm.

**Figure 7 sensors-25-02822-f007:**
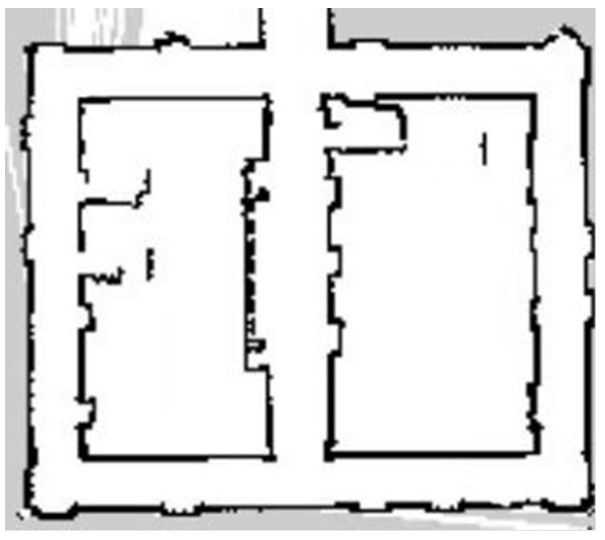
Indoor long corridor map constructed by improved gmapping algorithm in this paper.

**Figure 8 sensors-25-02822-f008:**
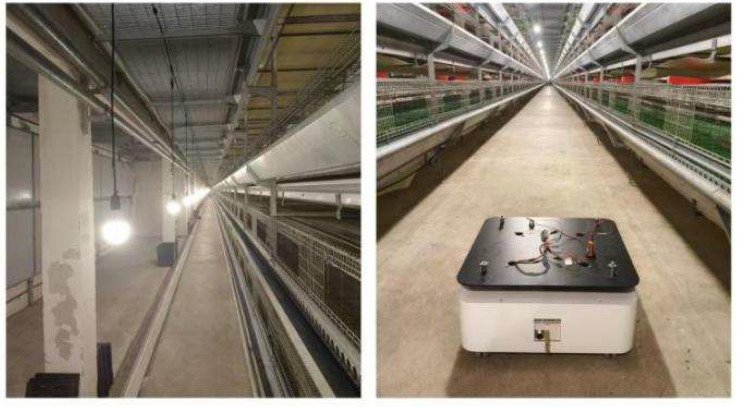
The actual environment of the chicken coop and the state of the mobile robot.

**Figure 9 sensors-25-02822-f009:**
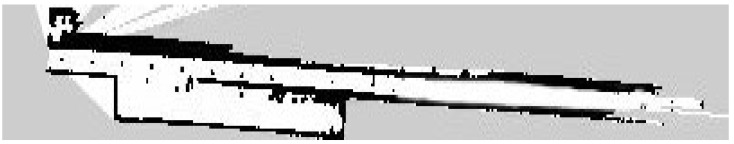
A single row of cage channel map constructed by traditional gmapping algorithm.

**Figure 10 sensors-25-02822-f010:**
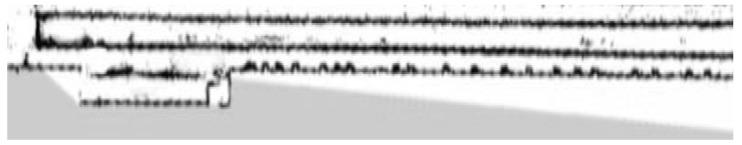
A single row of cage channel map constructed using the improved gmapping algorithm in this article.

**Figure 11 sensors-25-02822-f011:**
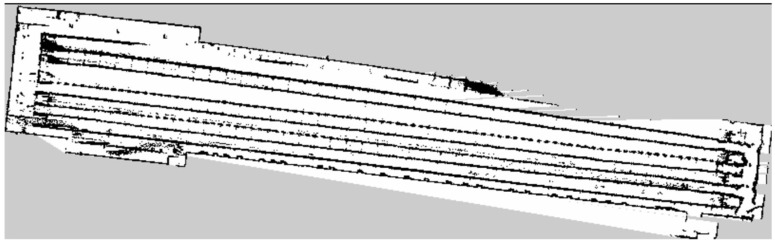
Complete chicken house environment map constructed by traditional gmapping algorithm.

**Figure 12 sensors-25-02822-f012:**
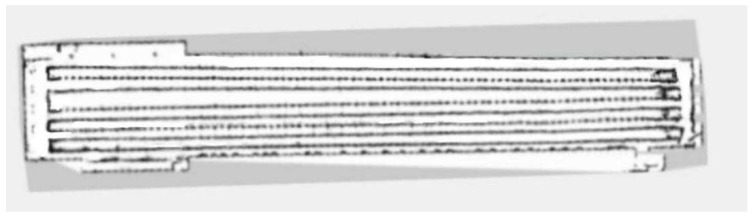
The complete chicken house environment map constructed by the improved gmapping algorithm in this paper.

**Figure 13 sensors-25-02822-f013:**
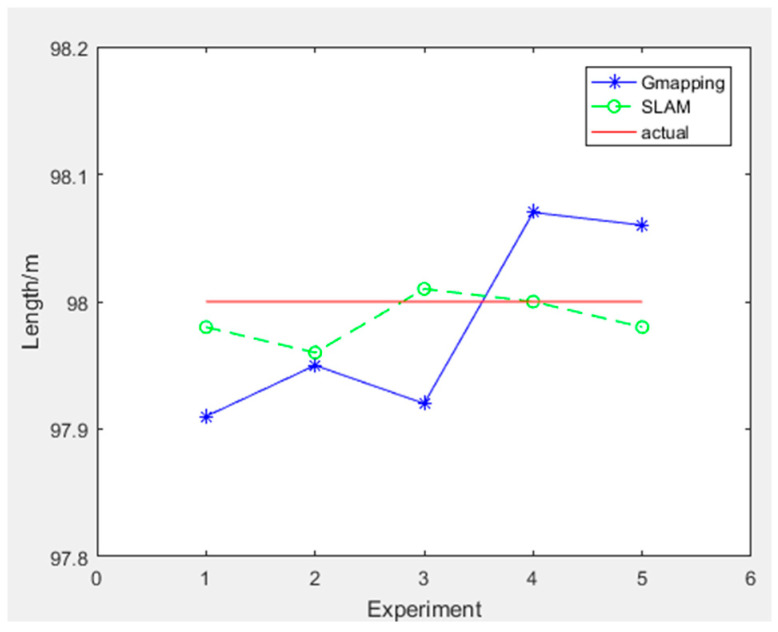
Chicken coop length measurement experiment results.

**Figure 14 sensors-25-02822-f014:**
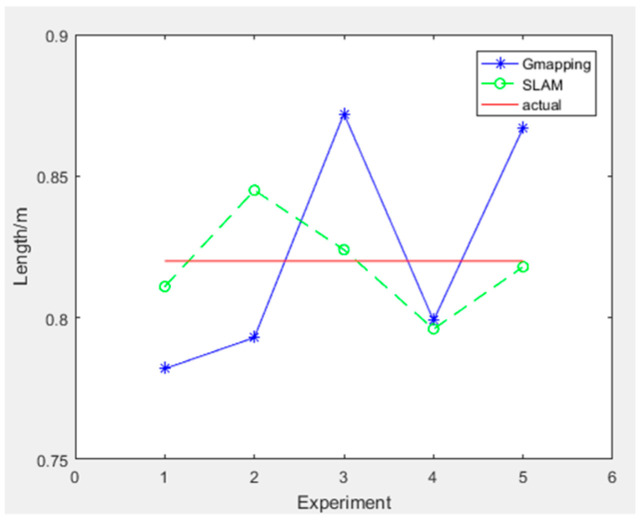
Distance between cages measurement experimental results.

**Figure 15 sensors-25-02822-f015:**
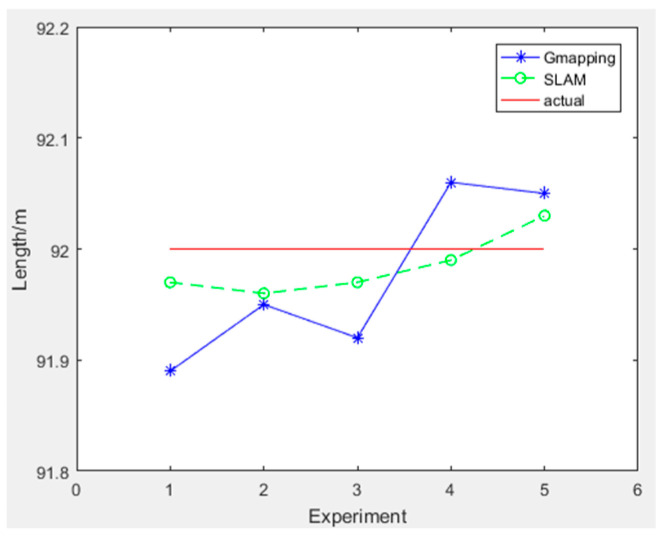
Cage length measurement experimental results.

**Table 1 sensors-25-02822-t001:** Comparison of LiDAR parameters.

Model	Velodyne-16	Mid-70	Mid-360
Distance measurement range	100 m	90–260 m	40 m
Range accuracy	±2 cm	±2 cm	±1 cm
Horizontal FOV	360°	70.4°	360°
Vertical FOV	30° (16 lines)	38.4° (64 lines)	59.2° (64 lines)
Angular resolution	2°	0.6°	0.925°
Weight	830 g	760 g	640 g
Power consumption	8 W	12 W	10 W

**Table 2 sensors-25-02822-t002:** Chicken coop length measurement experiment data.

Chicken Coop Length (98.000 m)	1	2	3	4	5	Mean Square Error	Maximum Absolute Error
traditional	97.910	97.952	97.922	98.074	98.068	0.0053	0.090
improved	97.981	97.966	98.017	98.005	97.985	0.0004	0.034

**Table 3 sensors-25-02822-t003:** Distance between cages measurement experiment data.

Distance Between Cages (0.820 m)	1	2	3	4	5	Mean Square Error	Maximum Absolute Error
traditional	0.782	0.793	0.872	0.799	0.867	0.0015	0.052
improved	0.811	0.845	0.824	0.796	0.818	0.0003	0.025

**Table 4 sensors-25-02822-t004:** Cage frame length measurement experiment data.

Cage Frame Length (92.000 m)	1	2	3	4	5	Mean Square Error	Maximum Absolute Error
traditional	91.895	91.953	91.928	92.062	92.051	0.0050	0.105
improved	91.976	91.968	91.970	91.993	92.034	0.0007	0.034

## Data Availability

Data are contained within the article.

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
