# Peer review of "Study on Optimization of Mapping Method for Multi-Layer Cage Chicken House Environment"

_sensors, 2025, doi:10.3390/s25092822_

Round 1
Reviewer 1 Report
Comments and Suggestions for Authors
This manuscript focuses on the problem of navigation map construction in livestock and poultry farming environments. The application background is clear, and the proposed improvements to the gmapping algorithm, especially in response to the complexity of multi-layer cage structures, are of practical significance. However, several issues must be addressed before the manuscript can be considered for publication.
There is a lack of detailed analysis regarding how different LiDAR models and their parameters affect the quality of the constructed maps.
The manuscript does not sufficiently analyze the sources of error under different environmental scenarios (e.g., are they due to ICP matching inaccuracies or LiDAR blind spots?).
The manuscript contains a large number of language issues, including Chinglish expressions and unclear logic. For example, the sentence "the ground is flat, but there are clustered obstacles, and some detailed features are not presented in the map" is logically contradictory.
The proposed improved method lacks quantitative comparisons with existing methods and does not provide convergence analysis or theoretical justification.
Comments on the Quality of English LanguageThe English could be improved to more clearly express the research.
Author Response
Dear Reviewer,
We sincerely appreciate your comments on improving the quality of this manuscript. Based on these suggestions, we have comprehensively revised the whole paper including grammar and some small mistakes, and we believe that it is much stronger as a result of the changes.
Comments 1: There is a lack of detailed analysis regarding how different LiDAR models and their parameters affect the quality of the constructed maps.
Response:
Thank you for reviewing our manuscript and providing valuable feedback. Regarding the "different LiDAR models" mentioned in the article, we realize that the expression was inappropriate and may have caused misconceive. We sincerely apologize for this.
We want to express the use of Mid-360 LiDAR to solve the problem. We have revised the wording. The specific modifications are highlighted in yellow on lines 98-100.
Comments 2: The manuscript does not sufficiently analyze the sources of error under different environmental scenarios (e.g., are they due to ICP matching inaccuracies or LiDAR blind spots?).
Response:
Thank you for pointing out the issue with the sources of error. The previous lines, 98-100, have been revised to use the Mid-360 laser radar to solve the mapping problem of chicken coops. The optimized gmapping algorithm has a better mapping effect with a maximum error of 3.4cm, and the measurement results are relatively stable. We believe that since the experiment was conducted with the same laser radar, the blind spot problem of the laser radar can be eliminated, reflecting the advantage of high mapping accuracy of the optimized gmapping algorithm. The specific modifications are highlighted in yellow on lines 98-100.
Comments 3: The manuscript contains a large number of language issues, including Chinglish expressions and unclear logic. For example, the sentence "the ground is flat, but there are clustered obstacles, and some detailed features are not presented in the map" is logically contradictory.
Response:
Thank you for pointing out the language issue. We have checked the wording of the article and made corrections to unclear and non-standard language expressions, clearly expressing the drawbacks of traditional gmapping methods and the advantages of improved gmapping methods. The specific modifications are highlighted in yellow on lines 307-310 and 326-432.
Comments 4: The proposed improved method lacks quantitative comparisons with existing methods and does not provide convergence analysis or theoretical justification.
Response:
Thank you for pointing out the issue with the experimental data. We have corrected the experimental data to a measurement accuracy of cm level, supplemented the quantitative analysis of the data, and specifically analyzed the high accuracy and stability of the optimization method compared to traditional methods. The specific modifications are highlighted in yellow on lines 362-405.

Reviewer 2 Report
Comments and Suggestions for Authors
The article is devoted to the modification of the algorithm of the robot used for disinfection of a chicken coop. The software, based on the information from the sensors, should perform mapping of the terrain and at the same time determine the robot's own location. The idea of ​​dividing this task into two subtasks is expressed in one of the cited literary sources. However, this idea is self-evident. The article develops this idea in order to increase the accuracy and reduce the computational complexity of the problems being solved.
Lines 364 - 373 provide comments on Table 1 and Figures 13, 14 and 15. These comments are insufficient. The reader should guess why there are several lines in the table and several points on the graph. What is this? Several identical experiments? Under the same conditions, or under different conditions?
If these are five different tests under identical conditions, then they should be processed as is customary in metrological testing, i.e. calculate and indicate the mathematical expectation and dispersion, as well as the standard deviation for a set of tests, this will be more illustrative. Than to offer a table with numbers, each having 8 significant digits.
Table 1 demonstrates the authors' carelessness in understanding the results they obtained. The fact is that no reality can be studied with an error of 0.000001, i.e. 0.00001%, except for very specific cases, such as absolute frequency measurements in spectroscopy. For this reason, the reader cannot trust such experimentally obtained numbers as, for example, 97.7831 or 97.9079. All experimental data in the table in columns 3, 4, 7 and 8 are given with an accuracy of six significant figures, given that the first figure is nine, that is, in fact, an accuracy of 0.00001% is stated. At the same time, the actual distance, which is given below in the table, is a value known with an accuracy of no more than four significant figures, that is, 0.01%. If you look at the obtained data, they differ in the second digit. Therefore, this type of experiment can hardly give data with an accuracy higher than 4-3 digits, and therefore such detailed numerical data should not be given, and instead of, for example, the number 97.7831, 97.8 ±0.1 should be given, or, perhaps, 97.78 ±0.01. From the illustrations it is clear that the spread of measurement results is approximately the same order of magnitude for the two compared methods, and it should be indicated what kind of accuracy is required for these applications? If the task at hand does not require high measurement accuracy, then the methods for increasing it are not relevant, and if the task requires a significant increase in accuracy, then the proposed methods are not effective enough, since the error still remains at the level of the same order of magnitude.
Overall, the article is quite good, but it is desirable to eliminate these shortcomings.
Author Response
Dear Reviewer,
We sincerely appreciate your comments on improving the quality of this manuscript. Based on these suggestions, we have comprehensively revised the whole paper including grammar and some small mistakes, and we believe that it is much stronger as a result of the changes.
Comments 1: Lines 364 - 373 provide comments on Table 1 and Figures 13, 14 and 15. These comments are insufficient. The reader should guess why there are several lines in the table and several points on the graph. What is this? Several identical experiments? Under the same conditions, or under different conditions?
Response:
Thank you for pointing out the issue with the unclear expression. The experiment described in lines 364-373 is a multiple-measurement experiment of three representative lengths on a map that has already been built in the chicken coop environment. Five measurement experiments were conducted for the chicken coop length, inter-cage distance, and cage frame length. Due to the complexity and ambiguity of the previous table, it was corrected to three tables, and the three representative data sets were quantitatively analyzed and presented clearly. The modification results are shown in Table 1, Table 2, and Table 3. The specific modifications are highlighted in yellow on lines 362-396.
Comments 2 & Comments 3: If these are five different tests under identical conditions, then they should be processed as is customary in metrological testing, i.e. calculate and indicate the mathematical expectation and dispersion, as well as the standard deviation for a set of tests, this will be more illustrative. Than to offer a table with numbers, each having 8 significant digits.
Table 1 demonstrates the authors' carelessness in understanding the results they obtained. The fact is that no reality can be studied with an error of 0.000001, i.e. 0.00001%, except for very specific cases, such as absolute frequency measurements in spectroscopy. For this reason, the reader cannot trust such experimentally obtained numbers as, for example, 97.7831 or 97.9079. All experimental data in the table in columns 3, 4, 7 and 8 are given with an accuracy of six significant figures, given that the first figure is nine, that is, in fact, an accuracy of 0.00001% is stated. At the same time, the actual distance, which is given below in the table, is a value known with an accuracy of no more than four significant figures, that is, 0.01%. If you look at the obtained data, they differ in the second digit. Therefore, this type of experiment can hardly give data with an accuracy higher than 4-3 digits, and therefore such detailed numerical data should not be given, and instead of, for example, the number 97.7831, 97.8 ±0.1 should be given, or, perhaps, 97.78 ±0.01. From the illustrations it is clear that the spread of measurement results is approximately the same order of magnitude for the two compared methods, and it should be indicated what kind of accuracy is required for these applications? If the task at hand does not require high measurement accuracy, then the methods for increasing it are not relevant, and if the task requires a significant increase in accuracy, then the proposed methods are not effective enough, since the error still remains at the level of the same order of magnitude.
Response 2 & Response 3:
Thank you for pointing out the issue with the experiment data. Previously, we did not fully consider data processing, so we corrected multiple sets of measurement data for quantitative analysis of the algorithm. The accuracy of LiDAR mapping is in cm, and the measurement data is in m. We take cm as the accurate data and m as the estimated data and keep the data to three decimal places. We have selected the maximum measurement error to reflect the measurement accuracy as the evaluation index and the mean square error to reflect the degree of dispersion of the measurement data. The specific modifications are highlighted in yellow on lines 376-405.

Round 2
Reviewer 1 Report
Comments and Suggestions for Authors
The authors have successfully addressed most of the previously raised concerns. The revised manuscript shows clear improvements in structure and experimental data. I would now recommend addressing the following minor issues:
Adding a brief discussion on how different LiDAR parameters (e.g., resolution, frequency, field of view) might affect mapping accuracy.
It is recommended that the authors enhance the Introduction section with additional related work to provide a more complete literature background. The following reference may be useful: 10.1016/j.psj.2024.104552
Consider including a brief theoretical explanation of the convergence or computational complexity of the improved ICP and classification-based resampling strategies, to strengthen the methodological rigor.
Author Response
Dear reviewer,
We sincerely appreciate your feedback on improving the quality of this manuscript. Based on these suggestions, we have carefully revised the entire paper, including rigor and material supplementation. We believe that the revised paper will be more powerful.
Comments 1: Adding a brief discussion on how different LiDAR parameters (e.g., resolution, frequency, field of view) might affect mapping accuracy.
Response:
Thank you for your suggestion on comparing and analyzing different parameters of LiDAR. We have added a comparative analysis of three laser radar parameters to clearly express the reason why we ultimately chose the Mid-360 laser radar. We mainly analyzed the measuring distance range and FOV. We found that the Mid-360 LiDAR is suitable for solving the mapping problem of chicken coop environments. The specific modifications are highlighted in yellow on lines 100-107.
Comments 2:It is recommended that the authors enhance the Introduction section with additional related work to provide a more complete literature background. The following reference may be useful: 10.1016/j.psj.2024.104552
Response:
Thank you for your suggestion on references. After carefully reading the references you recommended, we found that the egg detection method proposed in this article utilizes a classification weighting mechanism, which plays a significant role in enriching our research background. We have added it to our research background, which greatly enhances the completeness of the technical background of our article. Thank you for your careful review and valuable feedback.The specific modifications are highlighted in yellow on lines 70-72.
Comments 3: Consider including a brief theoretical explanation of the convergence or computational complexity of the improved ICP and classification-based resampling strategies, to strengthen the methodological rigor.
Response:
Thank you for your suggestion on Method rigor. To clearly express the advantages of our optimization method, we have corrected our language expression. We specifically analyzed the optimization methods and performance optimization analysis for improving the ICP algorithm. We specifically analyzed the performance optimization of classification resampling methods. The specific modifications are highlighted in yellow on lines 442-447.
